# Identifying Homogeneous Patterns of Injury in Paediatric Trauma Patients to Improve Risk-Adjusted Models of Mortality and Functional Outcomes

**DOI:** 10.3390/ijerph17030892

**Published:** 2020-01-31

**Authors:** Joanna F. Dipnall, Belinda J. Gabbe, Warwick J. Teague, Ben Beck

**Affiliations:** 1School of Public Health and Preventive Medicine, Monash University, Melbourne, Victoria 3004, Australia; belinda.gabbe@monash.edu (B.J.G.); Ben.Beck@monash.edu (B.B.); 2School of Medicine, Deakin University, Geelong, Victoria 3220, Australia; 3Health Data Research UK, Swansea University Medical School, Swansea University, Swansea SA2 8PP, UK; 4Trauma Service, The Royal Children’s Hospital, Melbourne, Victoria 3052, Australia; w.teague@unimelb.edu.au; 5Department of Paediatrics, University of Melbourne, Melbourne, Victoria 3052, Australia; 6Surgical Research Group, Murdoch Children’s Research Institute, Melbourne, Victoria 3052, Australia; 7Faculty of Medicine, Laval University, Quebec City, QC G1V 0A6, Canada

**Keywords:** latent class analysis, risk adjustment, paediatric, trauma, injury, mortality, KOSCHI, classes

## Abstract

Injury is a leading cause of morbidity and mortality in the paediatric population and exhibits complex injury patterns. This study aimed to identify homogeneous groups of paediatric major trauma patients based on their profile of injury for use in mortality and functional outcomes risk-adjusted models. Data were extracted from the population-based Victorian State Trauma Registry for patients aged 0–15 years, injured 2006–2016. Four Latent Class Analysis (LCA) models with/without covariates of age/sex tested up to six possible latent classes. Five risk-adjusted models of in-hospital mortality and 6-month functional outcomes incorporated a combination of Injury Severity Score (ISS), New ISS (NISS), and LCA classes. LCA models replicated the best log-likelihood and entropy > 0.8 for all models (N = 1281). Four latent injury classes were identified: isolated head; isolated abdominal organ; multi-trauma injuries, and other injuries. The best models, in terms of goodness of fit statistics and model diagnostics, included the LCA classes and NISS. The identification of isolated head, isolated abdominal, multi-trauma and other injuries as key latent paediatric injury classes highlights areas for emphasis in planning prevention initiatives and paediatric trauma system development. Future risk-adjusted paediatric injury models that include these injury classes with the NISS when evaluating mortality and functional outcomes is recommended.

## 1. Introduction

Injuries in children make a considerable contribution to disease burden globally, being a leading cause of mortality for children over one year of age and causing varying levels of disability affecting their development into adulthood [1]. A 10-year review of the injury outcomes of children in Australia found that injury was the leading cause of death in children aged 1 to 16 years [2]. The effects of rising globalisation, urbanisation, motorisation and environmental changes all impact on the risks and nature of childhood injuries around the world [3]. However, regardless of age or country, traumatic injuries can be debilitating, affect a variety of regions of the body, cause future physical and/or mental impairment, and can be fatal.

A single injury event in a child can result in highly variable patterns of injury and severity across age groups [2]. Furthermore, typical types of traumatic injuries are not static through childhood and adolescence, with the cause, pattern and severity of injury varying with age. For example, falls and burns are common in children aged 1 to 5 years as they explore their environment in the context of rapid growth and development [4], whereas the advent of risk-taking behaviour in the ascendency to adolescence introduces mechanisms of injury related to activities such as the use of bicycles [5,6] and off-road vehicles [7,8].

Many studies globally have focused on specific injury subsets (e.g., brain injury [9,10], spinal injuries [11], skeletal fractures [12], thoracic trauma [13]) or used single summary scores of severity such as the Injury Severity Score (ISS) [14] or New ISS (NISS) [15]. These approaches eliminate the ability to characterise patterns of injuries. Gaining a better understanding of the types and patterns of injuries sustained and how multiple injuries cluster together may allow for the improved modelling of injury mortality and functional outcomes.

Latent Class Analysis (LCA) measures one or more unobserved or latent classes inferred from a set of observed categorical variables [16]. This technique has been found to be superior to traditional cluster analysis, as LCA is a model-based approach where the selection of the number of classes is based on a set of statistical indices [17]. This approach has been widely used to create homogenous groups of individuals based on typologies in psychology, educational research and the social sciences [18].

Current risk-adjusted mortality models for paediatric trauma have predominantly used the ISS or NISS to control for the severity of the injury rather than pattern of injury. There is a need to focus on the individual paediatric trauma patient [19] and incorporate homogenous injury patient-centric groupings from LCA in these models. This would enable patterns of injuries to be taken into consideration. Therefore, this study aimed to identify homogeneous groups of paediatric major trauma patients for use in risk-adjusted models of mortality and functional outcome.

## 2. Materials and Methods 

### 2.1. Sample

This study used data from the Victorian State Trauma Registry (VSTR) [20]. The VSTR is a population-based registry containing pre-hospital, acute care and long-term outcomes data for all major trauma patients in Victoria, Australia. The Victorian State Trauma System defines ‘paediatric’ as aged 0 to 15 years, triaging the majority of paediatric major trauma patients to a single, designated paediatric Major Trauma Service (MTS) for definitive care [21]. Patients aged 0 to 15 years at the time of injury, injured between 2006 and 2016 (inclusive), with an ISS greater than 12 [22] were included in this study. The ISS ranges from 1 (least severe) to 75 (most severe) and an ISS>12 has been adopted to identify major trauma patients [22]. The VSTR has Human Research Ethics Committee approval from the Department of Health and Human Services (DHHS) for all 138 trauma-receiving hospitals in Victoria, and the Monash University Human Research Ethics Committee (MUHREC) (CF13/3040—2001000165).

### 2.2. Measures

Year of injury, demographics, injury event details, injury diagnoses, injury severity, and other relevant factors were extracted from the VSTR. Demographic data included sex (male, female) and age in years at the time of injury, categorised into four groups [23]: <1 year, 1–5 years, 6–10 years, and 11–15 years. The Australia Bureau of Statistics (ABS) Socio-Economic Indexes for Areas (SEIFA) were accessed, with the Accessibility/Remoteness Index of Australia (ARIA) (0 = regional, 1 = major city) and the quintiles for the Index of Relative Socioeconomic Advantage and Disadvantage (IRSAD) (low score = greater disadvantage, high score = greater advantage) were included as indicative measures of socioeconomic status and geographic remoteness. A binary variable was created to indicate if the patient was definitively managed at an MTS (0 = No, 1 = Yes).

Injury diagnosis codes were assigned by trained coders using the Abbreviated Injury Scale (AIS) (2005 version; 2008 update) [24]. The AIS is an internationally recognised tool for ranking injury severity, and classifies individual injury severity on a six-point scale (1 = minor injury to 6 = maximal (currently untreatable) injury). Sixteen dichotomous injury groups were identified (0 = no injury present; 1 = 1 or more injuries within group) based on the AIS body region and severity (Appendix A). The Injury Severity Score (ISS), New Injury Severity Score (NISS) and number of injuries per patient (i.e., across the sixteen injury groups) were calculated. The ISS is calculated as the sum of squares of the highest AIS code in each of six body regions (head/neck, face, chest, abdominal/pelvic contents, extremities/pelvic girdle, external). If an injury is assigned an AIS of currently untreatable injury, the ISS score is automatically assigned the highest ISS score of 75. The NISS is calculated from the sum of squares of the three highest AIS scores, irrespective of the body region affected.

Mechanism of injury was collapsed into 13 groups: motor vehicle occupant, motorcycle, cyclist, pedestrian, horse-related, low fall from a standing height or <1 m, high fall from ≥1 m, submersion/drowning, other threat to breathing, fire/scalds/contact burn, cutting, piercing object, struck by or collision with person/object, and “other” cause.

Binary measures of in-hospital mortality (1 = died, 0 = alive) and the Kings Outcome Score for Closed Head Injury (KOSCHI) scale [25] (1 = died in hospital/disability, 0 = good/intact recovery), administered at 6 months by telephone, were used for the final regression models.

### 2.3. Analyses

Patients were clustered by hospital to ensure standard errors allowed for intragroup correlation [26]. Individual hospital clusters of ≥10 patients were retained, with four clusters ranging in size from 17 to 1199 patients. Low volume hospitals (i.e., <10 patients) were classified into two clusters: metropolitan (n = 22) and regional (n = 29).

The main analyses consisted of six key steps, with steps 1 to 4 related to LCA and steps 5 to 6 related to the final regression models (Figure 1).

Four exploratory LCA Models were generated (M1, M2, M3, M4). To review the sensitivity of the final selection of the latent injury classes, two models excluded patients who sustained asphyxia or burn injuries (Ml and M3), and two models included patients with any asphyxia or burn injuries (M2 and M4). LCA M1 and M2 were initially analysed to establish the classes, then expanded to run the M3 and M4 multinomial logistic regression of the categorical latent variables on the covariates of age group and sex. Probabilities across models were compared to establish if these demographics influenced the latent injury class probabilities and latent classes (i.e., measurement known invariance explaining differences in class probabilities), or improved estimation [27]. Consultation with a paediatric trauma surgeon supported the final latent classes chosen.

The LCA took account of hospital clustering [28] and latent classes one to six were tested for each model to ensure an adequate number of classes were evaluated. Complex mixture modelling used maximum likelihood estimation with robust standard errors. Initial stage optimisations were set, and number of random sets of starting values for the final stage optimisation was set to one quarter of the initial starting values to ensure model estimation converged on the global maximum likelihood [29]. The maximum number of iterations in optimization was set to 20. 

Goodness of fit statistics were compared to establish the optimal number of classes: Akaike Information Criterion (AIC) [30], Bayesian Information Criterion (BIC) [31], Vuong-Lo-Mendell-Rubin Likelihood Ratio Test (LMR-LR) and adjusted LMR LR (ALMR-LR) test [32], and Entropy [33]. Smaller values of AIC/BIC and higher values for entropy indicated better model fit. The LMR-LR and ALMR-LR compared model fit improvement between models with k classes and (k − 1) classes. A *p*-value < 0.05 indicated rejecting the (k − 1) class model in favour of at least the current k class model. The LMR-LR and AMR-LR was applied to further classes to ensure a significant result for k + 1 classes (*p*-value > 0.05). The relative entropy criterion from MPlus was used for assessing the quality of class membership classification: 0.80 was considered high, 0.60 medium and 0.4 as low entropy [34].

Average latent class probabilities represented the proportion of the population expected to belong to a latent class. Estimated posterior probabilities ranged from 0 to 1, where higher posterior probabilities for injuries may indicate the label for that latent class.

Patients were classified into distinctive homogeneous groups, or latent classes, based on their posterior membership probabilities, given the model and the patient’s data. Relative proportions of latent classes established adequate proportions per class. Detailed descriptive analysis ensured each latent class was distinctly different, theoretically and substantively meaningful, and interpretable. Differences between the latent classes were explored using chi-square tests with Pearson adjusted residuals (AR) > |2| considered significant [35], and Kruskal–Wallis equality-of-populations rank tests with Dunn’s test [36] where appropriate.

Five multivariable logistic regression models were run to investigate the impact of the inclusion of the final latent injury classes on the risk-adjusted models for in-hospital mortality and 6-month functional outcome (KOSCHI) [37], controlling for potential demographic and injury confounders. The difference between the five models were inclusion of potential demographic and injury confounders and:Inclusion of ISS only (ISS)Inclusion of NISS only (NISS)Inclusion of final latent classes chosen only (LCA)Inclusion of ISS final latent classes chosen (ISS and LCA)Inclusion of NISS and final latent classes chosen (NISS and LCA)

The following post estimation goodness of fit statistics were used to evaluate each of the models:A BIC where the smaller the better model.A Hosmer–Lemeshow goodness-of-fit test using 10 quantiles where a *p*-value > 0.05 indicates a good model fit.Percentage sensitivity, specificity and overall percentage correctly classified where the higher the better.McFadden, Adjusted McFadden, McKelvey and Zavoina, Cox Snell, Nagelkerke R-square values where the higher the better.Receiver Operator Curve area (AUC) where the closer to 1 the better.Specification link test for single-equation models where a *p*-value > 0.05 indicates model correctly specified.

Mplus (Version 8.1) and R (Version 3.5.1) and associated packages [38,39], as well as Stata Version 15.1, were used for this research.

## 3. Results

### 3.1. Sample Characteristics

Of the 1285 patients meeting the inclusion criteria, four were excluded due to no injury profile classified from the AIS.

Males were predominant (66.5%), and the average age was 8 years (Table 1). Head injuries were the most common (brain injury and/or skull fracture 58.3%); 22.3% had sustained an isolated head injury. Falls, either low or high, were the most common injury mechanism (24.0%), followed by motor vehicle occupant (16.2%), then being struck by or colliding with a person or object (12.4%) (Table 2).

### 3.2. Latent Class Analysis

Fit information from the LCA were generally consistent across models. All models replicated the best log-likelihood. Based on a combination of the fit criterion and evaluation of the posterior probabilities, three latent classes were selected from M1 and M3; four latent classes were selected from M2 and M4. The largest decline in both AIC and BIC was between latent classes one to three for M1 and M3, and one to four for M2 and M4 (Figure 2). The BIC was lowest at latent classes three for all models; however, the four latent injury class models were selected for M2 and M4 as the BIC was virtually the same (i.e., <0.01% difference) but the entropy was higher and the classes made theoretical sense. The LMR-LR and ALMR-LR did not reach a significant result for the models, but the entropy values for all Models were above 0.8.

Figure 3 presents the prevalence of each latent injury class and the predicted probability that patients assigned to an injury class would have certain injuries for each model. Models 1 and 3 represented three latent classes: one was dominated by isolated head injuries (M1: 48.6%; M3: 49.4%); one represented multiple injuries (multi-trauma) (M1: 35.2%; M3: 34.4%); and one was dominated by isolated abdominal organ injuries (M1: 16.3%; M3: 16.2%). Three of the four latent injury classes for M2 and M4 were consistent with M1 and M3: isolated head injuries (M2: 45.3%; M4: 45.6%); multi-trauma injuries (M2: 29.0%; M4: 31.1%); and isolated abdominal organ injuries (M2: 15.2%; M4: 15.0%). Final latent injury classes consisted of the neck injuries, spinal injuries, vascular injuries, asphyxia and burns (M2: 10.4%; M4: 8.3%). For convenience, this class was labelled as ’other’ as it reflected the low numbers of paediatric major trauma patients with neck, spinal, vascular, and was dominated by asphyxia and burn injuries.

Exploratory analysis of the ARs for key characteristics across the final latent classes chosen found a number of consistent differences across the latent classes for the four models (Table 3). Patients associated with the isolated head injury latent class were more likely to be in the 0–5 years age group, discharged for rehabilitation, and had falls (low/high) or were struck by or collided with a person/object as their mechanism for injury. Patients associated with the multi-trauma latent class were more likely to be in the 11–15 years age group, discharged to rehabilitation and had been a motor vehicle occupant or pedestrian or riding a motorcycle as their mechanism for injury. Those associated with the isolated abdominal latent class were more likely to be male, in the 6–15 years age group, discharged directly to home and been a motorcyclist, cyclist or struck by or collision with person/object as their mechanism for injury.

### 3.3. Logistic Regression

#### 3.3.1. Mortality Models

The four latent injury classes from M4 were used for the mortality regression models. Patients with asphyxia injuries were separated out from the other latent class due to their threat to life differing from those patients with burns and the other injuries in this latent class. The results from the five logistic regression models for mortality indicated the best performing mortality model contained both the NISS and the modified latent classes, yielding the lowest BIC, a high sensitivity, specificity, overall correctly classified percentage and highest R-squared values and AUC (Table 4, Appendix B). In addition, the Hosmer–Lemeshow statistic indicated the model was a good fit, without specification error issues.

#### 3.3.2. Functional Outcome Models

As with the mortality modelling, the results from the five logistic regression models for the functional outcome models indicated the best performing functional outcomes model contained both the NISS and modified latent classes, yielding the lowest BIC, a high sensitivity, specificity, overall correctly classified percentage and the highest R-squared values for three of the four measures (Table 5, Appendix B). There was only a marginal difference in the AUC between models (4) and (5), but the Hosmer–Lemeshow statistic indicated model (5) was superior, without specification error issues.

## 4. Discussion

Irrespective of country or age, adequately characterising the number and complex pattern of injuries in major trauma patients can be challenging. This study used a novel methodological approach to classify key injury patterns across the whole paediatric major trauma population from Victorian registry data in Australia. For the first time, we are providing a quantified approach to succinctly describe injury patterns to the whole body, rather than relying on descriptions of injuries to individual body regions, and thereby providing data to directly inform injury prevention strategies. The use of exploratory LCA to identify three or four clusters provides an opportunity to target primary injury prevention strategies to prevent these specific injuries, and to focus strategies for optimizing the care of seriously injured paediatric trauma patients.

The identification of key injury patterns into the three main latent injury classes of isolated head injuries, isolated abdominal organ injuries and multi-trauma injuries was found to improve the model fit for both the mortality and 6-month functional outcome models. When considering childhood injury and the paediatric-specific trauma systems developed to prevent and care for such injury, the identification of ‘at risk’ injury populations, mechanisms and classes is of central importance. However, without accurate risk-adjustment, the design, application and evaluation of injury prevention strategies and trauma care quality improvements may be undermined by misleading epidemiological analyses. Therefore, we have proposed more accurate, paediatric-specific risk-adjustment modelling, to overcome these potential limitations, and so to promote more effective interventions for childhood injury, be that in prevention or trauma care delivery.

The findings highlight the role of isolated head and isolated abdominal injuries as common injury patterns in paediatric major trauma, elevating these for priority in the prevention and management of such injuries in childhood. Traumatic head injuries in children are a leading cause of death and a common cause of disability, often occurring in the very young [40]. The abdominal organ latent injury class—predominantly an isolated injury—highlights the importance and relevance of abdominal trauma in childhood injury. Children have relatively larger abdominal solid organs (e.g., spleen, liver and kidneys) compared to adults, which protrude below a more compliant (and so less protective) rib cage. These and other age-specific anatomical differences make children particularly vulnerable to abdominal organ injuries, such as those caused by bicycle handlebars [41]. 

The complex multi-trauma latent class is another important injury group in children, with approximately one in five patients in our data sustaining more than three injury types. These children exemplify the quality and complexity of injured children, for whom a systematic and paediatric-specific approach to trauma management has been associated with improved delivery and outcomes of care [4]. Not surprisingly, the multi-trauma latent class was significantly associated with higher energy mechanisms of injury including motor vehicle occupants, pedestrian collisions and motorcycle collisions. 

Previous studies of paediatric trauma have commonly used composite scores to describe the severity of injuries, e.g., ISS [4,21], while other studies have used the most severe injury [42] or the presence of specific (non-mutually exclusive) injury types [43]. This study identified patterns of injury in paediatric trauma to allow for a more informed and likely impactful modelling of injury in future paediatric trauma research. More parsimonious paediatric trauma statistical modelling was enabled by controlling for either three or four key latent injury classes, rather than each individual injury. The inclusion of injury classes in a model with NISS was superior in terms of all key fit statistics to the other four models. NISS has been shown to outperform the ISS for mortality in more severely injured adults [44], and this study showed that the NISS outperforms the ISS for mortality in more severely injured children. 

A strength of this research is the ability of our LCA modelling to contextualise the mortality or functional outcome expectations of an individual injured child in terms of homogenous injury groups. This overcomes an important and recognised limitation of previous injury research, in which the complexity of patterns of injuries sustained has limited inclusion in risk-adjusted statistical models. This research has focussed on the investigation of major trauma injury typologies in children, resulting in the simplification of the complexity of patterns of injuries. The sample size of the cohort used was considerably more than the sample size of 500 as recommended by Finch and Bronk [45] for LCA. A further strength is the improvement in the fit of the statistical models by the adjustment of key injury classes. 

A key limitation of this study is its focus on major trauma, given patterns of childhood injury are likely to differ in less-severely injured cohorts. This acknowledged bias notwithstanding, a mature trauma system aiming to reduce death and disability due to injury will similarly focus on patients with major trauma as the cohort of primary concern. Furthermore, the modelling presented in this study may have limited generalizability in non-Australian populations with dissimilar paediatric injury profiles, e.g., far higher rates of firearm trauma in the United States [46] or penetrating injury in the United Kingdom [47]. The initial classification of the sixteen injury groups used in the LCA relied on the broad dichotomous injury groupings denoting the presence or absence of one or more injuries in this cohort. The LCA technique used is an exploratory technique, assuming that latent classes do exist, with the number of classes defined prior to running the analysis. However, this study performed a number of LCA models to test for different number of classes, with and without covariates, using well-established fit statistics and theoretical understanding to decide the final number of classes. Since this is a retrospective study, future research involving a validation cohort from a developed country would be a worthwhile exercise for the evaluation of the performance of the models.

## 5. Conclusions

The key latent injury classes of isolated head, isolated abdominal organ and multi-trauma injuries revealed by this research are important for understanding the patterns of injuries sustained in paediatric major trauma, and will inform injury prevention and treatment strategies. It is recommended that researchers consider the inclusion of these injury classes with the NISS to refine future risk-adjusted paediatric injury models when evaluating mortality and functional outcomes.

## Figures and Tables

**Figure 1 ijerph-17-00892-f001:**
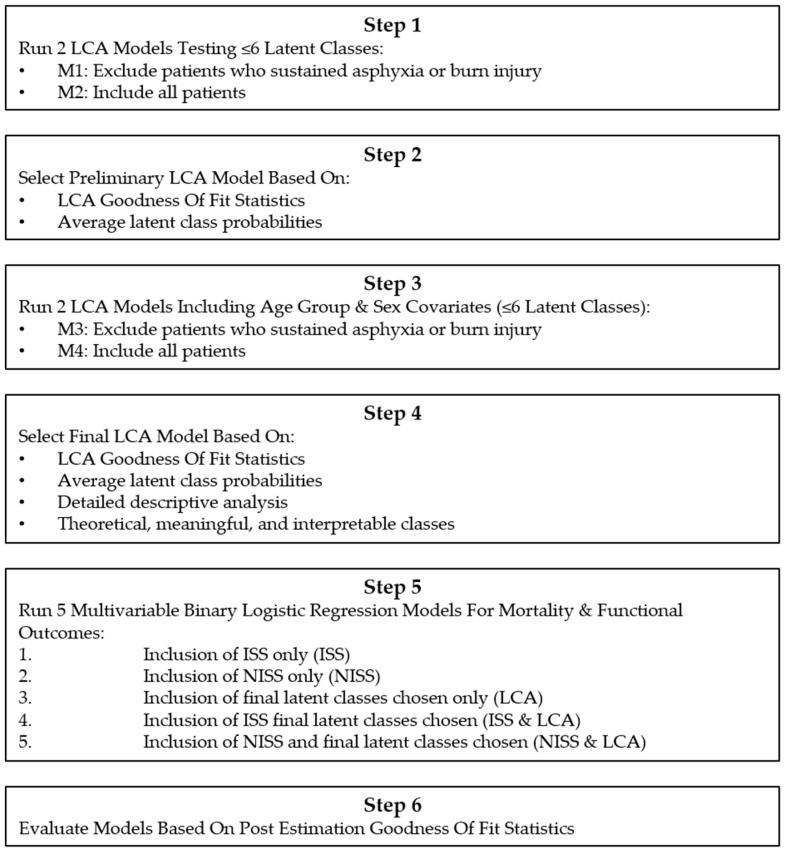
Flowchart of main analyses.

**Figure 2 ijerph-17-00892-f002:**
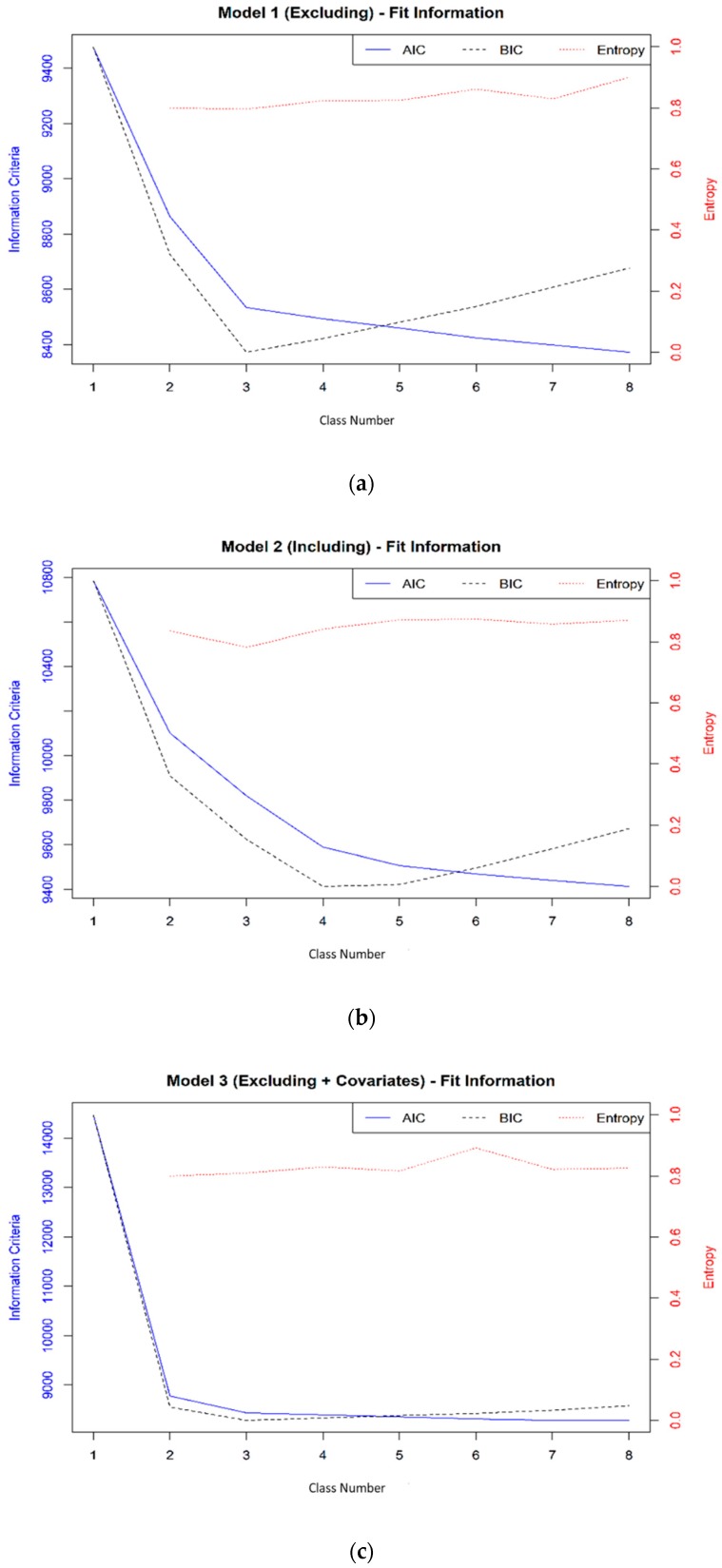
LCA Model’s AIC, BIC and Entropy. (**a**) Model 1, (**b**) Model 2, (**c**) Model 3, (**d**) Model 4.

**Figure 3 ijerph-17-00892-f003:**
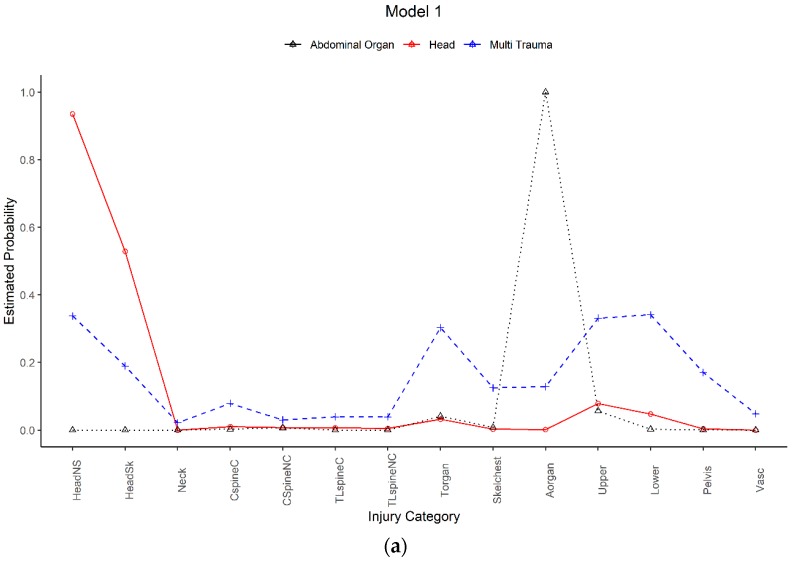
Estimated probabilities of Latent Classes (**a**) Model 1 (M1); (**b**) Model 2 (M2); (**c**) Model 3 (M3); (**d**) Model 4 (M4). Note: HeadNS = Head-brain injury, HeadSK = Head-skull fracture, Neck = Neck injuries, CspineC = Cervical spine–spinal cord injury (SCI), CSpineNC = Cervical spine–non-SCI, TLspineC = Thoracolumbar spine—SCI, TLspineNC = Thoracolumbar spine—non-SCI, Torgan = Thoracic organ injuries, Skelchest = Skeletal chest injuries, Aorgan = Abdominal organ injuries, Upper = Upper limb injuries, Lower = Lower limb injuries, Pelvis = Pelvis injuries, Vasc = Vascular injuries, Asph = Asphyxial injury, Burn = Burn injury.

**Table 1 ijerph-17-00892-t001:** Patient demographic characteristics.

Characteristic	Models 1 and 2	Models 3 and 4
***Base***	*1178*	*1281*
**Sex (n, %)**		
Male	790 (67.1%)	852 (66.5%)
Female	388 (32.9%)	429 (33.5%)
**Age Group (n, %)**		
<1 year	116 (9.8%)	122 (9.5%)
1–5 years	263 (22.3%)	326 (25.4%)
6–10 years	267 (22.7%)	278 (21.7%)
11–15 years	532 (45.2%)	555 (43.3%)
**Age in Years (Mean, SD)**	8.6 (5.2)	8.3 (5.2)
**IRSAD Quintile (n, %)**		
1	206 (17.7%)	230 (18.2%)
2	210 (18.0%)	227 (17.9%)
3	260 (22.3%)	281 (22.2%)
4	263 (22.6%)	291 (23.0%)
5	225 (19.3%)	236 (18.7%)
**ARIA (n, %)**		
Inner regional/outer regional/remote	394 (34.0%)	425 (33.8%)
Major city	764 (66.0%)	834 (66.2%)
**Fund (n, %)**		
Compensable	405 (34.7%)	410 (32.3%)
Non-compensable	762 (65.3%)	858 (67.7%)
**Major Trauma Service (n, %)**		
Yes	1028 (87.3%)	1112 (86.8%)
**Discharge Type (n, %)**		
Home	888 (75.4%)	940 (73.4%)
Rehabilitation	196 (16.6%)	202 (15.8%)
Hospital for Convalescence	18 (1.5%)	18 (1.4%)
Other	16 (1.4%)	20 (1.6%)
In-hospital death	60 (5.1%)	101 (7.9%)

**Table 2 ijerph-17-00892-t002:** Patient injury characteristics.

Characteristic	Models 1 and 2	Models 3 and 4
***Base***	*1178*	*1281*
**Injury Type (n, %)**		
Head—brain injury	503 (57.3%)	605 (52.8%)
Head—skull fracture	381 (32.3%)	381 (29.7%)
Neck injuries	9 (0.8%)	10 (0.8%)
Cervical spine—spinal cord injury (SCI)	39 (3.3%)	39 (3.0%)
Cervical spine—non-SCI	18 (1.5%)	18 (1.4%)
Thoracolumbar spine—SCI	20 (1.7%)	20 (1.6%)
Thoracolumbar spine—non-SCI	19 (1.6%)	19 (1.5%)
Thoracic organ injuries	152 (12.9%)	153 (11.9%)
Skeletal chest injuries	55 (4.7%)	55 (4.3%)
Abdominal organ injuries	246 (20.9%)	246 (19.2%)
Upper limb injuries	193 (16.4%)	193 (15.1%)
Lower limb injuries	169 (14.3%)	170 (13.3%)
Pelvis injuries	73 (6.2%)	74 (5.8%)
Vascular injuries	20 (1.7%)	20 (1.6%)
Asphyxial injury	Excluded	65 (5.1%)
Burn injury	Excluded	38 (3.0%)
**ISS (Median, IQR)**	17.0 (16.0, 26.0)	17.0 (16.0, 26.0)
**NISS (Median, IQR)**	25.0 (17.0, 34.0)	25.0 (17.0, 34.0)
**Six-Month KOSCHI**		
Death in hospital/disability	401 (57.4%)	463 (60.4%)
Good/intact recovery	298 (42.6%)	304 (39.6%)
**Total Injuries (Median, IQR)**	2.0 (1.0, 2.0)	1.0 (1.0, 2.0)
**Mechanism of Injury (n, %)**		
Motor vehicle occupant	207 (17.6%)	208 (16.2%)
Motorcycle	123 (10.4%)	124 (9.7%)
Cyclist	118 (10.0%)	118 (9.2%)
Pedestrian	143 (12.1%)	143 (11.2%)
Horse-related	40 (3.4%)	40 (3.1%)
Low fall	168 (14.3%)	168 (13.1%)
High fall	139 (11.8%)	139 (10.9%)
Submersion/drowning	Excluded	52 (4.1%)
Other threat to breathing	Excluded	12 (0.9%)
Fire/scalds/contact burn	Excluded	36 (2.8%)
Cutting, piercing object	13 (1.1%)	13 (1.0%)
Struck by or collision with person/object	159 (13.5%)	159 (12.4%)
Other	68 (5.8%)	69 (5.4%)

**Table 3 ijerph-17-00892-t003:** Significant Adjusted Residuals (AR) for key characteristics across final latent classes.

**Head Injuries**	**Model 1**	**Model 2**	**Model 3**	**Model 4**
Less than 1 year of age	6.58	6.66	7.68	8.20
1–5 years of age	4.92	2.08	5.36	2.37
Discharged for rehabilitation	2.19	3.26	-	2.51
Mechanism—low falls	8.27	9.22	8.18	9.25
Mechanism—high falls	4.28	5.34	4.37	5.36
Mechanism—struck by or collision with person/object	2.09	3.06	2.34	3.59
**Multi-Trauma**				
11–15 years of age	4.26	4.51	5.58	6.70
Discharged to rehabilitation	3.29	2.61	3.72	4.52
Mechanism—motor vehicle occupant	6.63	7.44	6.58	7.41
Mechanism—motorcycle	3.09	4.33	3.37	4.19
Mechanism—pedestrian	5.76	7.65	6.03	7.07
**Isolated abdominal**				
Male	2.84	3.03	2.79	3.03
6–10 years of age	2.67	3.19	2.72	3.00
11–15 years of age	3.93	4.41	4.00	4.56
Discharged directly home	7.23	7.77	7.43	7.77
Mechanism—motorcycle	4.31	4.78	4.34	4.78
Mechanism—cyclist	5.64	6.20	5.68	6.20
Mechanism—struck by or collision with person/object	2.01	2.54	2.04	2.54

Note: - denotes standardized AR < |2|.

**Table 4 ijerph-17-00892-t004:** Mortality logistic regression models.

	(1)	(2)	(3)	(4)	(5)
Model	ISS OR (s.e.)	NISS OR (s.e.)	LCA ^ OR (s.e.)	ISS and LCA OR (s.e.)	NISS and LCA OR (s.e.)
*Base*	1259	1259	1259	1259	1259
**ISS**	1.149 ***			1.161 ***	
	(0.007)			(0.008)	
**NISS**		1.111 ***			1.13 ***
		(0.005)			(0.009)
**LCA Class**					
Head			*Reference*	*Reference*	*Reference*
Multi-trauma			1.295 **	0.705 ***	2.474 ***
			(0.107)	(0.059)	(0.169)
Abdominal Organ			0.190 **	0.221	0.988
			(0.099)	(0.253)	(1.051)
Asphyxia			21.130 ***	29.957 ***	135.185 ***
			(4.921)	(5.895)	(30.330)
Burns and other			0.843	0.681	4.155 ***
			(0.149)	(0.328)	(1.678)
**Fit Statistics**					
BIC	481.93	471.55	546.90	411.30	**373.68**
Sensitivity %	85.86	**88.89**	76.77	83.84	**88.89**
Specificity%	82.07	82.41	78.45	87.59	**90.09**
Overall % correctly classified	82.37	82.92	78.32	87.29	**89.99**
Hosmer-Lemeshow goodness-of-fit test (*p*-value)	0.430	0.246	0.850	0.479	**0.743**
McFadden R^2^	0.357	0.372	0.263	0.458	**0.513**
Adjusted McFadden R^2^	0.342	0.357	0.248	0.444	**0.498**
McKelvey and Zavoina R^2^	0.507	0.529	0.398	0.558	**0.587**
Cox Snell R^2^	0.178	0.185	0.135	0.223	**0.246**
Nagelkerke R^2^	0.421	0.437	0.318	0.527	**0.581**
AUC	0.916	0.917	0.853	0.940	**0.951**
Specification link test (*p*-value)	<0.001 ***	0.029 **	0.139	0.077 *	**0.587**

OR = Odds Ratios, s.e. = Standard errors. *** *p* < 0.001, ** *p* < 0.05. ^ LCA modified to contain a separate group for Asphyxia injuries. Cut point for sensitivity %, specificity % and overall% correctly classified set to proportion died = 0.08.

**Table 5 ijerph-17-00892-t005:** Functional outcome logistic regression models

	(1)	(2)	(3)	(4)	(5)
Model	ISS OR (s.e.)	NISS OR (s.e.)	LCA ^ OR (s.e.)	ISS and LCA OR (s.e.)	NISS and LCA OR (s.e.)
*Base*	*743*	*743*	*743*	*743*	*743*
**ISS**	0.908 ***			0.918 ***	
	(0.012)			(0.010)	
**NISS**		0.940 ***			0.941 ***
		(0.008)			(0.006)
**LCA Class**					
Head			*Reference*	*Reference*	*Reference*
Multi-trauma			1.041	1.054	0.748 **
			(0.122)	(0.120)	(0.072)
Abdominal Organ			3.642 **	3.206 **	2.180*
			(1.830)	(1.533)	(1.007)
Asphyxia			0.179 ***	0.237 **	0.121 ***
			(0.079)	(0.101)	(0.042)
Burns and other			0.336 **	0.326 **	0.183 ***
			(0.126)	(0.122)	(0.054)
**Fit Statistics**					
BIC	847.90	846.47	858.12	792.31	**809.57**
Sensitivity %	73.83	71.81	74.83	**77.18**	76.51
Specificity %	66.29	**69.66**	64.94	68.09	**69.66**
Overall % correctly classified	69.31	70.52	68.91	71.74	**72.41**
Hosmer-Lemeshow goodness-of-fit test	0.358	0.931	0.768	0.369	**0.653**
McFadden R^2^	0.186	0.187	0.176	0.220	**0.224**
Adjusted McFadden R^2^	0.176	0.177	0.166	0.208	**0.214**
McKelvey and Zavoina R^2^	0.365	0.341	0.282	**0.392**	0.383
Cox Snell R^2^	0.221	0.223	0.211	0.257	**0.261**
Nagelkerke R^2^	0.299	0.301	0.285	0.347	**0.352**
AUC	0.777	0.782	0.775	**0.803**	0.806
Specification link test (*p*-value)	0.122	0.643	0.147	0.020 **	**0.150**

OR = Odds Ratios, s.e. = Standard errors. *** *p* < 0.001, ** *p* < 0.05, * *p* < 0.1. ^ LCA modified to contain a separate group for Asphyxia injuries. Cut point for sensitivity %, specificity % and overall% correctly classified set to proportion good/intact recovery = 0.40.

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
