# Peer review of "Identifying Homogeneous Patterns of Injury in Paediatric Trauma Patients to Improve Risk-Adjusted Models of Mortality and Functional Outcomes"

_ijerph, 2020, doi:10.3390/ijerph17030892_

Round 1
Reviewer 1 Report
Thank you for this interesting manuscript.
The results are important contributions to the research. However, it si sometimes difficult to know in the introduction and discussion if the data are associated with a local (Australia) or global context. It would be easier to compare the results with other context if the introdocution and discussion more stringent describe the researrch on a local and global level.
Author Response
Thank you for reviewing our manuscript and valuable feedback.
“The results are important contributions to the research. However, it si sometimes difficult to know in the introduction and discussion if the data are associated with a local (Australia) or global context. It would be easier to compare the results with other context if the introdocution and discussion more stringent describe the researrch on a local and global level.”
To address this comment, we have made some revisions to Section 1 Introduction and Section 4 Discussion of the manuscript:
The first paragraph of Section 1 Introduction in the manuscript has been expanded to highlight the global issues and similarities with Australia:
“Injuries in children make a considerable contribution to disease burden globally, being a leading cause of mortality for children over one year of age and causing varying levels of disability affecting their development into adulthood[1]. A 10-year review of the injury outcomes of children in Australia found that injury was the leading cause of death in children aged 1 to 16 years[2]. Effects of rising globalisation, urbaninsation, motorisation and environmental changes all impact on the risks and nature of childhood injuries around the world[3]. However, regardless of age or country, traumatic injuries can be debilitating, affect a variety of regions of the body, cause future physical and/or mental impairment, and can be fatal.”
This study relates to the classification of the anatomical characteristics of serious injuries in children, so in Section 4 Discussion of the manuscript we clarified this by the following sentence:
“Irrespective of country or age, adequately characterising the number and complex pattern of injuries in major trauma patients can be challenging. For the first time, this study used a novel methodological approach to classify key injury patterns across the whole paediatric major trauma population from Victorian registry data in Australia.”
Please note that the Discussion is focussed on our research predominantly relating to the classification injuries and the predictability of our models and ISS/NISS etc.
Reviewer 2 Report
Dear authors.
Thank you for letting me read your manuscript. I think it is interesting and useful. However, if you can present the materials and method section more clearly and coherent it would gain in validity. As it is now it is somewhat hard to follow your steps. Please clarify and argue about your choices. Also, I would like to read an ethical section.
Author Response
Thank you for reviewing our manuscript and valuable feedback.
“Thank you for letting me read your manuscript. I think it is interesting and useful. However, if you can present the materials and method section more clearly and coherent it would gain in validity. As it is now it is somewhat hard to follow your steps. Please clarify and argue about your choices.”
We have included a flowchart (Figure 1) to Section 2.3 of the manuscript outlining the steps in the analyses and alerting the reader to the rationale for choices which are expanded in the text.
“Also, I would like to read an ethical section.”
We have added the following ethics sentence to the paragraph to Section 2.1 of the manuscript:
“The VSTR has Human Research Ethics Committee approval from the Department of Health and Human Services (DHHS) for all 138 trauma-receiving hospitals in Victoria, and the Monash University Human Research Ethics Committee (MUHREC).”
Reviewer 3 Report
I would like to suggest to the Authors to make more clear the 3.2 chapter. And also to think about the discussion and what really could say "of new" this article because I really cannot see what is new in this research and so how could improve the injury prevention.
Author Response
Thank you for reviewing our manuscript and valuable feedback.
“I would like to suggest to the Authors to make more clear the 3.2 chapter."
We have added a flowchart (Figure 1) to Section 2.3 of the manuscript outlining the steps in the analyses to help clarify the order of results in Section 3.2 of the manuscript.
“And also to think about the discussion and what really could say "of new" this article because I really cannot see what is new in this research and so how could improve the injury prevention.”
Thank you for highlighting this point. We have emphasised the new and novel components of our research. This is the first time potentially complex injury combinations in paediatric major trauma patients has been classified and simplified. We have edited Section 4 Discussion to include the following sentences:
“Irrespective of country or age, adequately characterising the number and complex pattern of injuries in major trauma patients can be challenging. This study used a novel methodological approach to classify key injury patterns across the whole paediatric major trauma population from Victorian registry data in Australia. For the first time, we are providing a quantified approach to succinctly describe injury patterns to the whole body, rather than relying on descriptions of injuries to individual body regions, and thereby having the ability to directly inform injury prevention strategies. The use of exploratory LCA to identify three or four clusters provides an opportunity to target primary injury prevention strategies to prevent these specific injuries, and to focus strategies for optimizing the care of seriously injured paediatric trauma patients.
The identification of key injury patterns into the three main latent injury classes of isolated head injuries, isolated abdominal organ injuries and multi trauma injuries was found to improve the model fit for both the mortality and 6-month functional outcome models. When considering childhood injury and the paediatric-specific trauma systems developed to prevent and care for such injury, identification of ‘at risk’ injury populations, mechanisms and classes is of central importance. However, without accurate risk-adjustment, the design, application and evaluation of injury prevention strategies and trauma care quality improvements may be undermined by misleading epidemiological analyses. Therefore, we have proposed more accurate, paediatric-specific risk-adjustment modelling, to overcome these potential limitations, and so promote more effective interventions for childhood injury, be that in prevention or trauma care delivery.”
Reviewer 4 Report
Dear Editor and authors,
Dr. Dipnall et al. have conducted a population-based study to investigate the injury patterns in paediatric trauma. The topic and risk-adjusted models are interesting. I have some comments.
The authors found “The best models, in terms of goodness of fit statistics and model diagnostics, included the LCA classes and NISS.” (abstract section) However, the strength of ISS or NISS is easy to use for primary physicians. It provides a quick calculation and simple prognostic suggestion for primary care. It is appreciated to transform your LCA classes to plenary languages, integrate it into NISS, and educate the primary physicians how to apply them easily. It makes your model practical, feasible and beneficial for clinical practice.
Similarly, there are several interesting findings. “Identification of isolated head, isolated abdominal, multi-trauma and other injuries as key latent pediatric injury classes highlights areas for emphasis in planning prevention initiatives and paediatric trauma system development.” However, there is a gap to apply the models to clinical use. What’s your suggestions of “planning prevention initiatives and paediatric trauma system development”?
Why did you include ISS greater than 12? A brief introduction of ISS is appreciated.
Although binary measurements are convenient and the authors defined the 5 categories of KOSCHI outcomes to 2 categories (1=died in hospital/disability and 0=good/intact recovery). Further clarification is needed. Additionally, KOSCHI is designed for evaluation of traumatic brain injury, is it reasonable to apply it to damages without head injury, such as isolated abdominal injury?
In your analyses, did injury mechanism matter?
Child abuse is an important issue in childhood injury. How about the role of child abuse in present registry database?
Thank you very much!
Author Response
Thank you for reviewing our manuscript and valuable feedback.
“The authors found “The best models, in terms of goodness of fit statistics and model diagnostics, included the LCA classes and NISS.” (abstract section) However, the strength of ISS or NISS is easy to use for primary physicians. It provides a quick calculation and simple prognostic suggestion for primary care. It is appreciated to transform your LCA classes to plenary languages, integrate it into NISS, and educate the primary physicians how to apply them easily. It makes your model practical, feasible and beneficial for clinical practice.
Similarly, there are several interesting findings. “Identification of isolated head, isolated abdominal, multi-trauma and other injuries as key latent pediatric injury classes highlights areas for emphasis in planning prevention initiatives and paediatric trauma system development.”
Thank you for your positive comments and feedback. However, we would respectfully suggest that the ISS and NISS have little direct clinical relevance, and in prospect in the clinical care of injured children.
The method we have proposed enables a quick and easy understanding of the profile of injury. We are not trying to develop models for primary physicians. Rather, the intended application of our findings is in retrospectively describing patterns of childhood injury, and using these in more accurately risk-adjusted modelling.
“However, there is a gap to apply the models to clinical use. What’s your suggestions of “planning prevention initiatives and paediatric trauma system development”?”
We have responded to the Reviewer comments by adding the following to Section 4 Discussion of the Manuscript:
“When considering childhood injury and the paediatric-specific trauma systems developed to prevent and care for such injury, identification of ‘at risk’ injury populations, mechanisms and classes is of central importance. However, without accurate risk-adjustment, the design, application and evaluation of injury prevention strategies and trauma care quality improvements may be undermined by misleading epidemiological analyses. Therefore, we have proposed more accurate, paediatric-specific risk-adjustment modelling, to overcome these potential limitations, and so promote more effective interventions for childhood injury, be that in prevention or trauma care delivery. “
“Why did you include ISS greater than 12? A brief introduction of ISS is appreciated.”
The threshold of 12 was used based on a recent study by Palmer et al, 2016 where it was recommended an ISS >12 threshold be used to identify major trauma patients. (Palmer CS, Gabbe BJ, Cameron PA: Defining major trauma using the 2008 Abbreviated Injury Scale. Injury 2016, 47(1):109-115). This paper was referenced and we have clarified it further below.
The ISS was explained in some detail in Section 2.2. but we have moved some of this explanation into Section 2.1. and explained why we used ISS>12 in Section 2.1:
“The ISS ranges from 1 (least severe) to 75 (most severe) and an ISS>12 has been adopted to identify major trauma patients. “
“Although binary measurements are convenient and the authors defined the 5 categories of KOSCHI outcomes to 2 categories (1=died in hospital/disability and 0=good/intact recovery). Further clarification is needed. Additionally, KOSCHI is designed for evaluation of traumatic brain injury, is it reasonable to apply it to damages without head injury, such as isolated abdominal injury?”
The KOSCHI is considered a suitable functional outcome measure and a paediatric adaptation of the Glasgow Outcome Scale (GOS). Refer to Sleat et al, “Outcome measures in major trauma care: a review of current international trauma registry practice” Emerg Med J, 2011. We have included this reference in Section 2.2.
“Five multivariable logistic regression models were run to investigate the impact of the inclusion of the final latent injury classes on the risk-adjusted models for in-hospital mortality and 6-month functional outcome (KOSCHI)(Sleat et al, 2011)….”
“In your analyses, did injury mechanism matter?”
The focus of this paper was to investigate if there are latent injury classes and if the inclusion of these classes improved risk-adjusted models. For this reason, injury mechanism was adjusted for in the regression models, but not included in the primary discussion. Appendix B provides the details of the final models, including the standard errors and significance levels for injury mechanisms.
“Child abuse is an important issue in childhood injury. How about the role of child abuse in present registry database?”
Thank you for this question. The number of cases in the VSTR specifically identified child abuse as the intent was <4%. Future research could benefit from implementing this manuscript’s research strategies to potentially improve risk-adjusted models associated with this important issue.
Reviewer 5 Report
To:
Editorial Board
International Journal of Environmental Research and Public Health
Title: “Identifying homogeneous patterns of injury in paediatric trauma patients to improve risk-adjusted models of mortality and functional outcomes”
Dear Editor,
I read this manuscript and I think that:
The retrospective nature of this paper is a limitation of the study design. This should be discussed in a dedicated limitation section. Please provide. A validation cohort would be interesting for the evaluation of the performance of the model. Please discuss such a point. The role of care manager in such a setting can contribute to the best performance of the model as well as happens in cardiovascular settings (Ciccone MM et al. Vasc Health Risk Manag. 2010 May 6;6:297-305). Please discuss such a point.
Author Response
Thank you for reviewing our manuscript and valuable feedback.
“Dear Editor,
I read this manuscript and I think that:
The retrospective nature of this paper is a limitation of the study design. This should be discussed in a dedicated limitation section. Please provide. A validation cohort would be interesting for the evaluation of the performance of the model. Please discuss such a point.
We have added the following sentence to the limitations paragraph in Section 4 Discussion of the manuscript:
“Since this is a retrospective study, future research involving a validation cohort from a developed country would be a worthwhile exercise for the evaluation of the performance of the models.”
The role of care manager in such a setting can contribute to the best performance of the model as well as happens in cardiovascular settings (Ciccone MM et al. Vasc Health Risk Manag. 2010 May 6;6:297-305). Please discuss such a point.”
The role positive of care coordination in trauma care delivery is well recognised, with increasing attention being shown to the importance of providing models of care which simultaneously recognise and coordinate care for the injured and their family. Select references on this topic include Curtis K et al. Journal of Paediatrics and Child Health 52 (2016) 832–836, and Curtis K et al. Injury (2006) 37(7): 626-632. It is agreed that accurate risk-adjustment may inform the paradigm, delivery and evaluation of trauma care coordination. However, this would be no different from the informative role risk-adjustment in many other laudable aspects systematic paediatric trauma care, including prevention. Therefore, the importance of care coordination in trauma notwithstanding, the Authors respectfully suggest this is not a key point for discussion in this manuscript.
Round 2
Reviewer 1 Report
The authors have replied on all comments, and they have improved the manuscript. I think it is suitable for publication.
Reviewer 5 Report
The paper improved very much. The authors well addressed all previous comments.